# Breast Cancer Subtype-Specific miRNAs: Networks, Impacts, and the Potential for Intervention

**DOI:** 10.3390/biomedicines10030651

**Published:** 2022-03-11

**Authors:** Raj Pranap Arun, Hannah F. Cahill, Paola Marcato

**Affiliations:** 1Department of Pathology, Dalhousie University, Halifax, NS B3H 4R2, Canada; rpranaparun@dal.ca (R.P.A.); hannah.cahill@dal.ca (H.F.C.); 2Department of Microbiology & Immunology, Dalhousie University, Halifax, NS B3H 4R2, Canada; 3Nova Scotia Health Authority, Halifax, NS B3H 2YN, Canada

**Keywords:** microRNA (miRNA), breast cancer, subtype specificity, triple negative breast cancer (TNBC), human epidermal growth factor receptor 2 (HER2), estrogen receptor (ER), progesterone receptor (PR), prognosis

## Abstract

The regulatory and functional roles of non-coding RNAs are increasingly demonstrated as critical in cancer. Among non-coding RNAs, microRNAs (miRNAs) are the most well-studied with direct regulation of biological signals through post-transcriptional repression of mRNAs. Like the transcriptome, which varies between tissue type and disease condition, the miRNA landscape is also similarly altered and shows disease-specific changes. The importance of individual tumor-promoting or suppressing miRNAs is well documented in breast cancer; however, the implications of miRNA networks is less defined. Some evidence suggests that breast cancer subtype-specific cellular effects are influenced by distinct miRNAs and a comprehensive network of subtype-specific miRNAs and mRNAs would allow us to better understand breast cancer signaling. In this review, we discuss the altered miRNA landscape in the context of breast cancer and propose that breast cancer subtypes have distinct miRNA dysregulation. Further, given that miRNAs can be used as diagnostic and/or prognostic biomarkers, their impact as novel targets for subtype-specific therapy is also possible and suggest important implications for subtype-specific miRNAs.

## 1. Introduction

Three decades ago, the first microRNA (miRNA), lin-4, was identified in *Caenorhabditis elegans* [1]. The seminal paper was the first to suggest the regulation of a protein (LIN-14) through complementary sequence interaction of the miRNA with the 3′untranslated region (UTR) of the lin-14 mRNA. Subsequent research has demonstrated the universal importance of miRNAs, which is reflected by their ubiquitous presence and abundance across all genera from viruses to mammals. Calin et al., first described a potential miRNA role in cancer; miR-15 and miR-16 were deleted in chromosome 13 in chronic lymphocytic leukaemia [2]. In the past two decades, miRNAs found to play critical roles in the regulation of cancer pathways have been labelled oncomirs and tumor suppressor mirs based on their prominent oncogenic and tumor suppressive effect [3]. Notable examples include miR-21, miR-17~92 and the miR-200 family [4].

Over 2000 miRNAs have been identified in humans [5]. In the genome, miRNA genes are intragenic or intergenic, and some miRNA genes are clustered into a polycistronic transcription unit which are expressed together in multiple conditions [6]. Micro RNAs have an established naming convention where they are named by sequential numbers, and if the same miRNAs are generated from distinct genetic locations, they are identified by hyphenated numbers (e.g., miR-10-1), whereas closely related mature miRNAs are indicated by a letter suffix (e.g., miR-200 family members; miR-200a, miR200b, and miR200c). Mature miRNAs are labelled with the 5p or 3p suffix to denote their origin arm of the stem loop [7].

Post-transcription, mature functional miRNAs are generated via a multi-step process. The primary pri-miRNA transcripts consist of single or multiple loop sections and single-stranded fragments, which are cleaved by the microprocessor complex involving RNaseIII Drosha and its cofactor subunit DiGeorge syndrome critical region 8 (DGCR8) in the nucleus, resulting in pre-miRNAs [6]. The Drosha and DGCR8 microprocessor complex binds to the stem loop section and cleaves at 11 bp from base of stem loop [8,9]. Post-export to the cytoplasm through the exportin5, the loop part is cleaved by Dicer to form duplex RNA, each strand ~20 nucleotide long. Dicer recognizes the single nucleotide overhangs of the stem loop pri-miRNA, and the Dicer–RNA complex is stabilized by the trans-activation-responsive RNA binding protein (TARBP2) [3,10]. This duplex RNA is bound by Argonaute protein AGO to form RNA induced silencing complex (RISC) and generate the mature miRNAs [11]. One of the mature miRNA strands (passenger strand) is degraded by the nuclease activity of AGO2, while the other strand (guide strand) is incorporated into RISC, and subsequently regulates post-transcriptional gene expression via binding to complementary sequences in mRNA targets. This typically results in the downregulation of target mRNAs by inhibiting their translation and/or mediating their degradation via RISC [4,5,7,8,9]. In specific conditions of serum or amino acid starvation and in quiescent cells, miRNA mediated upregulation of translation is also observed [12,13].

The process of post-transcriptional gene regulation by miRNAs is essential in the regulation of all biological processes including development and differentiation, immune responses, cell cycle progression, cell death, stress responses, and metabolism [14,15,16,17,18,19,20]. The essentiality of miRNAs is also reflected in cancer, where dysregulation of miRNAs is commonly paired with cancer development, progression, and treatment responses [21]. For breast cancer, dysregulation of miRNA happens early in the development of ductal carcinoma [22]. In the seminal 2005 paper, Iorio et al. described distinct miRNA signatures between normal and breast cancer cells with significant deregulation of miR-125b, miR-145, miR-21 and miR-155, which are mostly known for their tumor suppressive function [23]. Further, in breast cancer specifically, the importance of miRNAs is evident not only in the progression of the disease, but also in the specific breast subtypes which define the disease clinically and molecularly and is the focus of this review.

## 2. Breast Cancer Subtypes Defined by Gene and miRNA Expression

Breast cancer is one of the most common malignancies, with multiple subtypes, based on clinical parameters and molecular profiling. In addition to disease staging, the expression status of hormone receptors’ estrogen receptor (ER), progesterone receptor (PR), and human epidermal growth factor receptor 2 (HER2) in tumors define the prognosis of the cancer and treatment options [24,25,26]. Hence, clinically, breast cancers are defined as ER+/PR+, HER+, or triple-negative (i.e., those lacking expression of these three receptors). This classification system allows for the administration of endocrine therapies in the hormone expressing subtypes. Additionally, breast cancer subtyping is observed via transcriptome profiling, which has identified four major subtypes (luminal A, luminal B, HER2, and basal-like). The ER+/PR+ breast cancers are predominately luminal A/B and TNBCs are predominately basal-like. Clearly gene expression defines breast cancer [27]; it is then not surprising that the expression of miRNAs also displays subtype-specificity.

With the increasing availability of microarray and small RNA sequencing technologies, significant advancement has been possible in identifying global changes in miRNA dysregulation. Meta-analysis of patient sample repositories has also proven a versatile tool for the identification and study of miRNAs involved in breast cancer. Clustering analysis of all patient tumors based on miRNA expression show separation based on breast cancer subtype, demonstrating that miRNAs affect subtype phenotypes [22,28,29].

Most of the studies included in this manuscript are from patient cohorts that identify important miRNAs among specific patients; these studies compare cancer tissue with normal tissue to identify the cancer specific miRNAs. By comparing the enriched miRNAs among different breast cancer subtypes, the subtype specific miRNA signature and common breast cancer miRNAs are identified. Direct relative expression of miRNAs among different subtypes are studied in in vitro experiments, where the controlled environment allows for the comparison between two cancer subtypes instead of comparing against a standard control.

This variation in the abundance of specific miRNAs associated with specific subtypes and the potential influence this has on subtype-associated gene expression, signaling and clinical outcome is discussed here for each major subtype and summarized in Table 1.

### 2.1. ER+/PR+ Breast Cancers

Estrogen and progesterone hormones influence normal breast development and in breast cancer they play important roles in the disease progression, which is governed by the presence of estrogen and progesterone receptors. The luminal subtypes consist predominately of ER+/PR+ breast cancers and comprise the majority of breast cancers [25]. ER-associated miRNAs are enriched in the luminal subtypes. A key distinguisher between the luminal A and B subtypes is the expression of the Ki67 proliferation marker; luminal A breast cancers have low Ki67, while luminal B breast cancers have high Ki67 [46]. The subtype-specific expression of Ki67 reflects the better prognosis and low tumor grade associated with the luminal A subtype, and the comparably worse prognosis and higher tumor grade associated with the luminal B subtype. Furthermore, in terms of miRNA dysregulations, we see distinguishing features in the two luminal subtypes. Clustering analysis including breast cancers of all subtypes showed that luminal A and luminal B are clustered close to each other but were distinct from one another [22]. Further, there is more prominent dysregulation in miRNAs in luminal B vs. luminal A breast cancers. Specifically, 657 miRNAs were found dysregulated in luminal B cancers and 67 miRNAs were dysregulated in luminal A cancers. Among the luminal-dysregulation miRNAs, miR-1290 is significantly reduced in luminal A Ki67 low tumors [35]. MiR-1290 is oncogenic in a context driven nature [47]; its reduction in luminal A correlates with its clinical outcome.

The distinct luminal A-miRNA signature is defined by miR-30c-5p, miR-30b-5p, and miR-99a/let-7c/miR-125b cluster, while the luminal B subtype is enriched with miR-182-5p, miR-200b-3p, miR-15b-3p, miR-149-5p, miR-193b-3p and miR-342-3p, 5p [22,30,32,33,34]. Haakensen et al. linked miR-30b-5p, miR-30c-5p, miR182-5p and miR-200b-3p to better prognosis in patients with luminal subtype breast cancer [22]. Given the prominence of miR-30 miRNAs among luminal A breast cancers, it important to note that miR-30 is a cluster of six miRNAs (miR-30a, -30b, 30-c1, -30c-2, -30d, 30e), and as whole the miR-30 family generally inhibit migration and growth [48] (Figure 1). For example, miR-29b/miR-30d regulate migration through lysyl oxidase-like 4 (LOXL4) inhibition [49]. Early evidence demonstrated that miR-30 inhibit cell division through cyclin D2 targeting [50], and inhibition of c-Myc-induced carcinogenesis [51]. The other prominent luminal A-associated miRNA cluster, miR-99a (miR-100)/let-7c/miR-125b), is also typically paired with tumor suppression and reducing growth rate and migration in breast cancer (Figure 1). This cluster reduces tumor growth by inhibiting proteins involved in important cellular processes, including homeobox A1 (HOXA1), mammalian target of rapamycin (mTOR), insulin-like growth factor binding protein 1 (IGFBP1), and fibroblast growth factor receptor 3 (FGFR3) [52,53,54,55]. The miR-99a cluster also inhibits oncogenes, including Yamaguchi sarcoma viral oncogene homolog 1 (YES1), ETS proto-oncogene 1 (ETS1) and ETS variant transcription factor 6 (ETV6) [23].

The increased expression of miRNAs associated with tumor suppression among luminal A breast cancers is consistent with the relatively slow growth of luminal A breast cancers; however, there is also evidence that in advanced cancer stages, miR-30 plays an oncogenic role. In patients with advanced tumors, miR-30b-5p is highly expressed in tissue and circulation [38], suggesting a potential change in function towards tumor progression. This is an important caveat when considering miRNA function in cancers; it is always context dependent and reliant on the abundance of relative target mRNAs, which is also shifting during cancer progression. The function of miRNAs is never in isolation and dependent upon the evolving transcriptome.

Additionally, differences in circulating levels of miR-29a, miR-181a, and miR-652 is also evident in the serum samples of luminal breast cancers. All three miRNAs are significantly downregulated in both the tumors and serum of patients with luminal A vs. luminal B breast cancers [33]. Treatment regimens also influence circulating miRNA levels, with tamoxifen treatment of ER+ tumors resulting in increased miR-221 in serum [31].

### 2.2. HER2 Overexpressing Breast Cancers

HER2 is a tyrosine kinase receptor that belongs to the family of epidermal growth factor receptors (EGFR). HER2 overexpression in breast cancers is present about 15–20% of breast tumors, and like TNBC, it is associated with worse patient prognosis and survival [56]. However, unlike TNBCs, HER2+ (i.e., HER2 overexpression) in breast cancers can be targeted. Paired with chemotherapy, the monoclonal antibody against HER2, trastuzumab, is used to treat HER2+ breast cancers. However, resistance and recurrence are common, so other drugs have been developed such as the tyrosine kinase inhibitor lapatinib and the monoclonal antibody pertuzumab, which prevents HER2 dimerization and signaling [57]. HER2 overexpression results in specific gene signatures in breast cancer, especially in the cancer-promoting pathways. The increased mitogen-activated protein kinase (MAPK), phosphatidyl-inositol-3 kinase (PI3K)/AKT and HER3 receptor signaling [58,59], characteristic of HER2+ breast cancers, result in the downstream effects on enhanced cell proliferation and the observed aggressive clinical phenotype.

HER2+ breast cancers also have a distinct miRNA expression profile. Lowery et al. profiled 453 miRNAs in 29 tumors and found that that the HER2+ subtype is associated with a number of miRNAs, including miR-302c, miR-520d, miR-181c, miR-376b and miR-30e [37]. Additionally, in a study that evaluated 221 breast cancer tumors and 49 normal tissue controls, miR-125b is reportedly upregulated in HER2+ breast cancers [36]. MiR-4728-3p expression is also associated with HER2+ breast cancers and its gene is encoded within a HER2 intron [60]. Other miRNAs identified in HER2+ breast cancers include upregulation of miR-21 and miR-146a-5p, while miR-181d and miR-195-5p are downregulated [61]. These altered miRNA expression profiles are specific to the HER2+ subtype and could be exploited in diagnostic/prognostic tools.

In terms of a connection with cancer phenotypes associated with the HER2+ breast cancer subtype, these miRNAs also have function beyond just expression associations. For example, miR-125b is specifically connected to metastasis of HER2+ breast cancers [62] and with worse patient outcomes [39]. However, the effects of the miRNA in cancer are again context dependent, as miR-125 has also been reported to have tumor suppressive effects in various cancers [63]. For HER2+ associated miR-4728-3p, its mRNA targets include downstream targets of HER2 signal transduction and the estrogen receptor alpha (ESR1) [64]. Recent investigation into miR-4728 indicate that when it is overexpressed in HER2-postive tumors, the efficacy of HER2 inhibitor laptinib is minimized (Figure 1) [65]. This was linked to decreased expression of pro-apoptotic NOXA59. Further, miR-4728-3p mediates stabilization of miR-21-5p in HER2+ breast cancer (Figure 1) [66], facilitating miR-21-5p mediated oncogenesis [67,68].

As demonstrated with miR-4728 [65], the effect of miRNA on drug treatments are not uncommon, likely because miRNAs are expressed in a context dependent manner. Identifying changes to the miRNA landscape before and after treatment will aid in the development of improved treatments for HER2+ breast cancers. Normann et al. utilized four HER2+ breast cancer cell lines to assess whether treatment with trastuzumab and lapatinib together or separately change the miRNA landscape [57]. Levels of miRNAs hsa-let-7b, miR-1236, miR-134, miR-25, miR-3656, miR-3663-3p, miR-3940 and miR-885-5p were significantly altered following drug treatment [57]. Importantly, treatment with miRNA mimics (e.g., miR-101-5p mimic) sensitized cells to treatment with the drugs. Additionally, miR-101-5p downregulates HER2 and MAPK1; its expression is associated with higher survival rates in HER2+ tumors. These findings together indicate that miR-101-5p acts as a tumor suppressor in HER2+ cancers.

### 2.3. TNBC

TNBC is one of the major contributors of breast cancer mortality; nearly 25% of the overall breast cancer-related deaths are among patients with TNBC, despite representing only 10–15% of breast cancers [40]. TNBCs are aggressive in terms of proliferation, with a high mitotic index and high Ki67 staining in histology. They also recur more frequently than other breast cancer subtypes. Like the other above reviewed subtypes, TNBC also has reported correlations with specific miRNAs. For example, survival analysis of patients reveal correlation with multiple miRNAs in TNBC. Specifically, miR-27a/b, miR-210 and miR-30e are associated with worse survival, and miR-155 and miR-493 are associated with better survival in TNBC [41,42,43]. Further, miR-374a/b and miR-454 are associated with disease free survival [64]. Some miRNAs are reported to be associated with metabolic processes in TNBC; miR-210, miR-105-5p and miR-767-5p are essential for the Warburg effect, with miR-210 involved in glucose uptake, lactate production and extracellular acidification rate in TNBC [43].

TNBCs exhibit significant heterogeneity, both phenotypically and genotypically [44,69]. Hence, TNBCs can be further subdivided into multiple subtypes. Based on gene expression profiling, TNBC is divided into six major subtypes; basal-like 1 (BL1), basal-like 2 (BL2) immunomodulatory (IM), claudin-low mesenchymal (M), mesenchymal-stem like (MSL) and luminal androgen receptor positive (LAR), and each of these have unique clinical outcomes, phenotypes and drug sensitivities. For example, the two mesenchymal TNBC subtypes are associated with epithelial–mesenchymal–transition (EMT) gene signatures and pronounced migratory capacity. The individual miRNA signature for molecular subtypes has not been studied in detail for most of the TNBC molecular subtypes. However, significant information on basal-like TNBC are achieved through in silico analysis of patient cohort studies with miRNA studies. The molecular influences of some of the miRNAs are studied in detail in in vitro experiments detailing its function in TNBC.

For example, Rinaldis et al. identified a distinct four miRNA signature in TNBC and demonstrated that the miR-17~92 and miR-106b-25 clusters are significantly overexpressed in basal-like TNBCs [70]. It is noteworthy that the miR-17~92 cluster is of particular importance in the TNBC transcriptome and are among the most well-studied miRNAs (Figure 1). The miR-17~92 cluster comprises of miR-17, miR-18a, miR-19a, miR-20a, miR-19b-1 and miR-92a-1; together these miRNAs regulate expression of transcription factor E2F1, thrombospondin1 (THBS1), connective tissue growth factor (CTGF) and phosphate and tensin homolog (PTEN) [71,72,73]. Earlier, O’Donnell et al. showed that the proto-oncogene cMYC modulates the critical transcription factor E2F1 by regulating the miR-17~92 cluster, resulting in cancer proliferation [74]. This intricate oncogene signaling regulation has earned the miR-17~92 cluster the moniker oncomiR-1. Apart from involvement in cMYC oncogene signaling, the oncomiR-1 cluster also inhibits inositol polyphosphate-4-phosphatase type II B (INPP4B) and is distinctly associated with the BL1TNBC subtype [75]. Notably, INPP4B is a known inhibitor of the PI3K/AKT mediated growth pathway; knockdown experiments clearly showed the tumor suppressive role of INPP4B [76]. Further, Kalecky et al. found distinct differences between BL1 and BL2; miR17~92 cluster, miR-17, miR-18a and miR-19a were high in BL1 tumors, but not as high in BL2 tumors [75].

The BL1 subtype of TNBC is associated with low PTEN and overall decreased miRNA expression [77]. This group is also associated with worst survival among TNBCs. Recent evidence on structural mapping of miR-17~92 cluster showed a sub-optimal micro processing of primary miRNAs, resulting in unequal expression of the constituent miRNAs of the cluster [78]. This highlights the potential role of regulatory molecules of miRNA processing in the regulation of the miR-17~92 cluster [79,80]. This cluster may play a crucial role in the subtype characteristics associated with BL1-TNBCs and the distinction of TNBCs from ER+/PR+ and HER2+ breast cancers.

Like the miR-17~92 cluster, miR-135b shows molecular subtype specificity among TNBCs; miR-135b is upregulated in TNBCs in general but is most overexpressed in BL1 and BL2 TNBC subtypes [81]. Importantly, miR-135b regulates expression of the ER, androgen receptor (AR) and hypoxia inducible factor 1 alpha subunit inhibitor (HIF1AN) [82]; thereby, its expression in TNBCs may directly affect hormone receptor loss and contribute to this distinguishing feature of TNBCs. Interestingly, the AR significantly reduces miR-135b expression [81]. Further, miR-135-b also targets large tumor suppressor kinase 2 (LATS2), a intermediator of Hippo signaling, and promotes cell proliferation in unstratified breast cancer patient tumors [83]. In TNBC, miR-135-b targets adenomatosus polyposis coli (APC), a WNT signaling regulator [84]. Both these targets are important regulators of migratory process. These studies observed increased proliferation and migration in the cells treated with miR-135b mimics. However, in early-stage breast cancer, miR-135b-5p repression resulted in migration through the syndecan-binding protein (SDCBP) [85]. These findings strongly suggest that miR-135-5p has a role in the regulation of migration processes in breast cancer, with functional difference among different subtypes arising from context-specific signaling networks.

Currently, most of the TNBC molecular subtype data with respect to miRNA associations is predominately with the BL1 and BL2 subtypes. Given the importance of miRNA in gene expression regulation, it is likely that there are specific miRNAs associated with the other molecular subtypes; however, they just remain unstudied. The MSL and M molecular subtypes exhibit migratory phenotypes with pronounced expression of EMT genes. These genes are repressed in the basal-like cells; hence, the likelihood of a role for miRNA-mediated regulation is high in distinguishing these phenotypes.

Although the associations with specific molecular subtypes was not addressed in a study that compared miRNA expression among different breast cancer cell lines, Adams et al. reported downregulation of miR-34a and miR-200 in TNBCs compared to ER+ breast cancer and non-tumorigenic breast cell lines (Figure 1) [86,87]. These specific miRNAs are connected with EMT regulation in breast and other tumors, with effects in migration [88,89,90,91]. Inhibition of miR-34a promotes the migratory phenotype in MDA-MB-231 cells through the IL6 pathway [88]. Notably, MDA-MB-231 is classified as a mesenchymal TNBC cell line, and miR-34a is downregulated in the cell line [88].

The miR-200 family consists of miR-141, miR-200a, 200b, 200c and miR-429 [92]. It is one of the most well-studied miRNA families with extensive effects in the regulation of migration, via its multiple members and targets. Notably, inhibitors or antagomirs of miR-200a inhibited critical EMT genes N-cadherin, Snail and Twist [93], and other genes involved in migration, like proto-oncogene Jun, ETS1, brain-specific homeobox/POU domain protein 3A (BRN3) and zinc finger E-box binding homeobox 1 (ZEB1), were regulated by all miR-200 family members [93]. The miR-200 family associations with some TNBC cell lines may be DNA methylation dependent. Interestingly, TNBC cell lines with mesenchymal phenotypes, like MDA-MB-231 cells and lung metastasis sub-clones of MDA-MB-468 cells (MDA-MB-468 LN), have hypermethylated promoters of the miR-200 family [94,95]. Demethylation of the miR-200c promoter site induced its expression and reduced cell migration [96]. ZEB1 facilitates this epigenetic regulation of the miR-200 family [97].

### 2.4. Therapeutics of miRNA in Breast Cancer

The importance of miRNAs in breast cancer suggests they have therapeutic potential, and this could be achieved by either depleting oncogenic miRNAs or enriching tumor suppressive miRNAs [98,99]. Elimination of oncogenic miRNAs is possible by delivering an oligomer complementary to the target miRNA. These antagomirs bind to mature miRNAs, resulting in inhibition and degradation of the target miRNAs [98]. Enriching tumor suppressive miRNAs is done by delivering miRNA mimics (double stranded RNA sequences that have the same sequence as the miRNA) in cells; importantly, with mimics they are considered native to the cell [100]. By comparing published work on miRNA-based therapeutics, Krützfeldt et al. shortlisted twelve miRNAs being studied with therapeutic interest in breast cancers [98]. Three miRNAs in particular, miR-21, miR-10-b and miR-34a, have been extensively studied with profound preclinical therapeutic potential, with both antimetastatic and anti-proliferative properties. Reintroduction of miR-34a in TNBCs reduces both migration and proliferation in cell culture [86] and in vivo [101]. Combination therapy of doxorubicin and miR-34a delivered using nanoparticles was effective in TNBCs models [102]. Among antagomir studies on breast cancer, miR-21 and miR-10 antagomirs has shown promise in breast cancer [103,104,105]. It is important to note that both miR-21 and miR-34a appear to have critical influence on cancer signaling in specific breast cancer subtypes; hence, it is prudent to test the therapeutic or diagnostic/prognostic role of miRNAs in cell lines that have been clearly subtyped. Cell line classifications based on breast cancer subtypes have been summarized and should be carefully considered [106,107].

## 3. Conclusions

In this review, we have highlighted both the seminal papers in the field and the recent findings demonstrating miRNAs associations among certain breast cancer subtypes. We have focused our discussion on the miRNA examples that have functional consequences in the breast cancer subtypes. We have delineated important miRNA-initiated regulation of gene expression that leads to cancer phenotypes that are prominent within specific breast cancer subtypes. Importantly, the transcriptome network can be “enriched” in certain tumor-promoting or tumor suppressive phenotypes based on the abundance of certain miRNAs and their targets. Eventually these dynamics lead to effects on clinical fates. Furthermore, it is not one miRNA in isolation but the intricate signaling network of multiple miRNAs and their many gene targets that will result in specific tumor or subtype characteristics. We also reviewed the emerging evidence for a role in miRNAs in the molecular subtypes of TNBCs; this area needs further investigation.

Clearly, miRNAs play a role in the progression of breast cancer and the eventual patient outcomes. This can be exploited or intervened in new therapeutic strategies for breast cancer. Further analysis on the role of specific miRNAs and novel agents for manipulation of tumor-specific miRNAs is required. Likely, any effective intervention may require the consideration of multiple miRNAs or miRNA networks and be specifically targeted towards patients with tumors expressing the required miRNA-target profile. This review adds value to the importance of miRNAs in breast cancer subtypes and has showcased the need for further study.

## Figures and Tables

**Figure 1 biomedicines-10-00651-f001:**
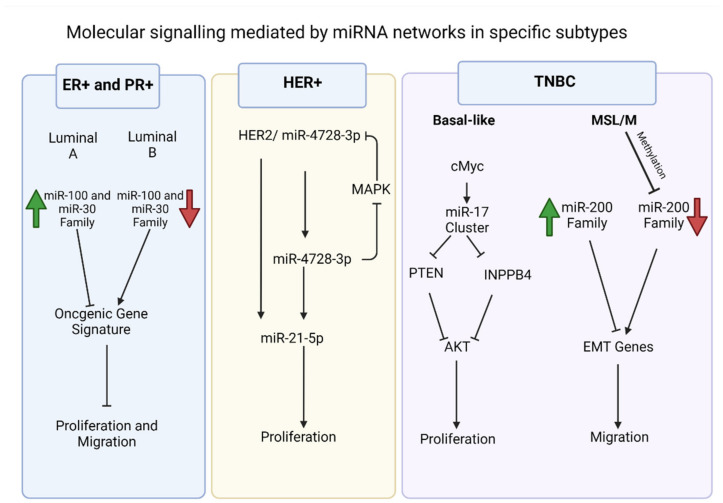
Examples of significant microRNAs (miRNAs) associated with specific breast cancer subtypes and their effects on cell phenotypes. In estrogen receptor positive/progesterone receptor positive (ER+/PR+) breast cancers (blue box), we note the role of miR-100 and miR-30 families and the distinction between the luminal A and luminal B molecular subtypes. In human epidermal growth factor receptor 2 positive (HER2+) breast cancers (yellow box), we have noted miR-4728-3p, which is present in the intronic region of HER2 and is co-expressed with HER2. The miRNA is involved in feedback regulation of HER2 and oncogenic miR-21-5p. This facilitates several oncogenic processes in later stage tumors. In triple negative breast cancers (TNBCs) (purple box), we have noted the cMYC oncogene driven miR-17~92 cluster, which is overexpressed in TNBCs, specifically the BL1 molecular subtype. The miRNA cluster promotes proliferation through its direct targets, which include phosphate and tensin homolog (PTEN) and inositol polyphosphate-4-phosphatase type II B (INPP4B) and are inhibitors of the proliferation mediator AKT. The migratory phenotype of TNBC cell lines that fall within the mesenchymal stem like (MSL) and mesenchymal (M) molecular subtypes is facilitated by inhibition of the miR-200 family through epigenetic changes that allows expression of epithelial to mesenchymal transition (EMT) and migration genes, resulting in the migratory phenotype.

**Table 1 biomedicines-10-00651-t001:** Distinct microRNA (miRNA) signatures and individual miRNAs associated with specific breast cancer subtypes. Breast cancer gene (BRCA), estrogen receptor (ER), progesterone receptor (PR), triple negative breast cancer (TNBC), human epidermal growth factor receptor 2 (HER2), quantitative real time polymerase chain reaction (qPCR), The Cancer Genome Atlas (TCGA), gene expression omnibus (GEO), formalin-fixed paraffin-embedded (FFPE).

Analysis Type	Findings	miRNA Signature	Subtype	Reference
microRNA (miRNA)quantitative real-time polymerase chain reaction (qPCR) panel on 139 breast cancer (BRCA) patient tissues compared against 26 normal tissues	patients with miR-182-5p and miR-200b-3p expression show better prognosis	miR-30c-5p, miR-30b-5p, miR-182-5p, and miR-200b-3p	ER/PR +	[30]
case control study of estrogen receptor/progesterone receptor positive (ER/PR +) patients with tamoxifen treatment	miR-221 expression is high in ER/PR + patients and is not changed by KI67/PR levels	miR-221	ER/PR +	[31]
clustering analysis of RNA seq from patient cohort of 186 patients	miR-99a/let-7c/miR-125b cluster is high in luminal A compared to luminal B	miR-99a/let-7c/ miR-125b	ER/PR +	[32]
qPCR of 54 luminal A type patients against 56 controls	study identified miRNAs specifically downregulated in luminal A type patients	miR-29a, miR-625, miR-181a	ER/PR +	[33]
miRNA qPCR of luminal A patients compared against controls	diagnostic markers for luminal A	miR-145, miR-195 and miR-486	ER/PR+	[34]
immunohistochemistry of miR-1290 targets among 256 ER positive breast cancer	miR-1290 is a prognostic marker for luminal breast cancers	miR-1290	ER/PR+	[35]
meta-analysis of patient datasets	specific miRNA signature between luminal A and luminal B breast cancer subtypes	miR-30b-5p, miR-30c-5p high in luminal A, miR-182-5p, miR-200b-3p, miR-15b-3p, miR-149-5p, miR-193b-3p and miR-342-3p, 5p high in luminal B	ER/PR+	[22]
mimic transfection, luciferase activity and qPCR	miR-125b functions as a competitive endogenous RNA with EPOR and ERBB2	miR-125b	HER2+	[36]
in silico and qPCR analysis of 300 miRNAs	upregulated miRNA biomarkers for human epidermal growth factor receptor 2 (HER2) subtype	miR-146a-5p	HER2+	[37]
in silico and qPCR analysis of 300 miRNAs	downregulated miRNA biomarkers for HER2 subtype	miR-181d and miR-195-5p	HER2+	[38]
miRNA screen (1626) in combination with targeted treatments lapatinib and trastuzumab	tumor suppressive miRNA signature identified, treatment with mimics sensitize cells to trastuzumab and lapatinib	miR-101-5p, mir-518a-5p, miR-19b-2-5p, miR-1237-3p, miR-29a-3p, miR-29c-3p, miR-106a-5p, and miR-744-3p	HER2+	[39]
protein expression and The Cancer Genome Atlas (TCGA) data analysis	overexpression of miR-4728 in HER2 minimizes the effect of laptinib	miR-4728		
triple negative breast cancer (TNBC) vs. non-TNBC patient samples	diagnostic markers of TNBCs	hsa-miR-10a, hsa-miR-18a, hsa-miR-135b and hsa-miR-577	TNBC	[40]
miR arrays from stored tissues	distinct in TNBCs compared to ER negative patients	miR-10a, miR-18a, miR-135b and miR-577	TNBC	[41]
miR arrays from stored tissues	basal-like subtype has overexpression of both clusters, derived from copy number	miR-17-92 and miR-106b-25 cluster overexpression	TNBC	[42]
	upregulated in TNBCs, prognostic signature	miR-455-3p, miR-107, miR-146b-5p, miR-17-5p, miR-324-5p, miR-20a-5p and miR-142-3p,		
	downregulated in TNBCs	miR-139-5p, miR-10b-5p, miR-486-5p		
regression analysis of patient data compared against gene expression omnibus (GEO) datasets	upregulated in TNBCs	miR-455-3p, miR-107, miR-146b-5p, miR-17-5p, miR-324-5p, miR-20a-5p and miR-142-3p	TNBC	[43]
	clustering of miRNAs significantly different between TNBCs and HER2+ subtypes	miR-139-5p, miR-10b-5p, miR-486-5p		
meta-analysis of published research articles	upregulated in TNBCs, prognostic signature	miR-10b, miR-21, miR-29, miR-9, miR-221/222, miR-373	TNBC	[44]
	downregulated in TNBCs	miR-145, miR-199a-5p, miR-200 family, miR-203, miR-205		
miRNA extraction and microarray from formalin-fixed paraffin-embedded (FFPE) tissues	TNBC-specific four miRNA signature which is reduced in other subtypes	miR-17-5p, miR-20a-5p, miR-92a-3p, miR-106b-5p	TNBC	[45]

## Data Availability

Not applicable.

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
