# Peer review of "Breast Cancer Subtype-Specific miRNAs: Networks, Impacts, and the Potential for Intervention"

_biomedicines, 2022, doi:10.3390/biomedicines10030651_

Round 1
Reviewer 1 Report
This review summarize the different miRNA signatures specific for the different breast cancer subtypes. It is useful and sound to highlight these differences.
However, some revision are needed in order to improve the quality.
- Both in the title and in the abstract there is the reference to the potential use of miRNA differently expressed between breast cancer subtypes as therapeutics. However, I could not find in the main text any references to this aspect. Please add some reference to this aspect.
- It is not clear, throughout the whole text, whether the specific subtype miRNA signatures have been identified compared to control group or other breast cancer subtypes. It is important to specify this aspect as a signature specific for e.g. luminal A found comparing a control group might not be useful to discriminate between different subtypes.
- It would be useful to add some information on the different cell lines referring to their subtype classifications, as it was done for MDA-MB-231 in the TNBC paragraph, also for the other subtypes. As this review is well structured and informative on the differences on BC subtypes, it is important for scientists to report the specific differences to the cell lines most commonly used in the lab.
- Sentence from line 171 to line 174 it is said that miRNAs are “differentially expressed” without specifying whether they are upregulated or downregulated in HER2+BC. Could you please indicate it?
- In table 1 it is better to add a last column with the list of references, rather than insert the references in the findings column
- There are some sentences that need grammar and spell check: line 82-83, line 124-126, line 195-198, line 315
- Sentence from line 130 to line 136 is too long. Please divide it in two sentences.
- The last sentence of the conclusion (line 328) is truncated. You should conclude it.
- I would suggest giving a final reading of the entire manuscript before submission. I might have missed some other sentences that need spell check.
Author Response
We would like to thank both the reviewers for their critical comments and suggested corrections and inclusions for the manuscript. The suggested corrections have improved the manuscript. We have answered all the reviewer comments and the modified sections in the manuscript, which are highlighted.
We respond to each comment below:
- Both in the title and in the abstract there is the reference to the potential use of miRNA differently expressed between breast cancer subtypes as therapeutics. However, I could not find in the main text any references to this aspect. Please add some reference to this aspect.
- We thank the reviewer for this observation. The manuscript was lacking in this aspect. We have corrected this and included a section in the review on the context of miRNAs in therapeutics among breast cancer subtypes. Please refer to lines 317-334 for the included section on miRNAs in therapeutics for breast cancer.
- It is not clear, throughout the whole text, whether the specific subtype miRNA signatures have been identified compared to control group or other breast cancer subtypes. It is important to specify this aspect as a signature specific for e.g. luminal A found comparing a control group might not be useful to discriminate between different subtypes.
- We appreciate this comment, this is a hurdle in many miRNA studies and we suggest that there is a need for more direct comparisons between subtypes. Many of the studies referenced in this manuscript involve patient cohort studies and compare either cancer patients to normal patients or compare a miRNA signature. For these studies, we specify that the work suggests expression in a subtype, not that the miRNAs are specific to that subtype only. More work needs to be done to discriminate this point.
- We have included some studies that do compare miRNAs among individual breast cancer subtypes.
- It would be useful to add some information on the different cell lines referring to their subtype classifications, as it was done for MDA-MB-231 in the TNBC paragraph, also for the other subtypes. As this review is well structured and informative on the differences on BC subtypes, it is important for scientists to report the specific differences to the cell lines most commonly used in the lab.
- Discussing the individual cell lines subtype lineage is beyond the scope of the current manuscript. This information is available in other reports, and we have added two references that describe and categorize breast cancer cell lines into subtypes (new references 106 and 107). The comment is appreciated, and we have described the importance of consideration of cell line subtype when designing experiments using miRNAs (see lines 331-334).
- Sentence from line 171 to line 174 it is said that miRNAs are “differentially expressed” without specifying whether they are upregulated or downregulated in HER2+BC. Could you please indicate it?
- We have corrected the ambiguous term “differentially expressed” and the specific expression is mentioned in the sentence.
- In table 1 it is better to add a last column with the list of references, rather than insert the references in the findings column
- We have added a separate column of references to the table in the corrected manuscript.
- There are some sentences that need grammar and spell check: line 82-83, line 124-126, line 195-198, line 315
- Thank you for the detailed reading and suggested language corrections. We have made necessary corrections in the corresponding lines suggested by the reviewer.
- Sentence from line 130 to line 136 is too long. Please divide it in two sentences.
- We have made changes to the sentence to make it more readable.
- The last sentence of the conclusion (line 328) is truncated. You should conclude it.
- We have rephrased the sentence to convey the same with better readability (new lines 351-353).
- I would suggest giving a final reading of the entire manuscript before submission. I might have missed some other sentences that need spell check.
- As suggested, we did a final read-through of the manuscript and made some further corrections in the corrected manuscript.
Reviewer 2 Report
The manuscript is properly prepared. The authors introduce and discuss the content related to the role of miRNAs in breast cancer in a concise but interesting way. References are well chosen.
For minor remarks, please correct the column on the role of miRNA in the table - the form of entering the information needs to be standardized. It seems to me that the most important role of miRNAs in monitoring therapy and in developing new therapeutic concepts is too poorly described. Could this section of the manuscript be extended?
Author Response
We would like to thank both the reviewers for their critical comments and suggested corrections and inclusions for the manuscript. The suggested corrections have improved the manuscript. We have answered all the reviewer comments and the modified sections in the manuscript, which are highlighted.
Below, we respond to each specific point:
For minor remarks, please correct the column on the role of miRNA in the table - the form of entering the information needs to be standardized. It seems to me that the most important role of miRNAs in monitoring therapy and in developing new therapeutic concepts is too poorly described. Could this section of the manuscript be extended?
- We appreciate the reviewer’s comments and have corrected the table and rephrased the findings/role section to be more uniform.
- We have also included a new section in the manuscript describing the role of miRNAs in breast cancer therapeutics, highlighting the most important ones.